# Chronic Hyponatremia: The Role of Reset Osmostat in Patients with Suspected SIAD

**DOI:** 10.3390/jcm13123538

**Published:** 2024-06-17

**Authors:** Aline Kiewiet, Ruben Schuinder, Joan Doornebal, Paul Groeneveld

**Affiliations:** 1Department of Internal Medicine, University Medical Center Groningen (UMCG), 9713 GZ Groningen, The Netherlands; r.j.schuinder@umcg.nl; 2Department of Internal Medicine, Isala, 8025 AB Zwolle, The Netherlands; j.doornebal@isala.nl (J.D.); p.h.p.groeneveld@isala.nl (P.G.)

**Keywords:** hyponatremia, reset osmostat, SIAD, water-diluting test, copeptin

## Abstract

**Background:** Hyponatremia is common, particularly among the elderly. Reset osmostat (RO) serves as an alternative diagnosis to the syndrome of inappropriate antidiuresis (SIAD). There is limited information available regarding the prevalence of RO in outpatient clinics and hospital wards. The water-diluting test is considered the gold standard for the diagnosis of RO. The recent identification of copeptin provides an additional diagnostic marker alongside the utilization of fractional uric acid excretion. **Methods:** This single-center, prospective, observational study involved eight patients undergoing a water-diluting test over a study period of 2 years. **Results:** Reset osmostat was diagnosed in 50% of cases, while SIAD was confirmed in one patient. The tests were inconclusive for the remaining three patients. **Conclusions:** Our findings suggest that reset osmostat, despite its rarity, is a plausible diagnosis in chronic hyponatremia. The relevance of copeptin could not be confirmed in this study. Moreover, fractional uric acid excretion might be as effective as the water-diluting test in diagnosing reset osmostat.

## 1. Introduction

Electrolyte imbalances, especially hyponatremia, are frequently found in patients, especially among the elderly. Hyponatremia, defined as a serum sodium concentration of less than 135 mmol/L [1], can be broadly categorized into disruptions in water or sodium homeostasis, or both. The predominant causes encompass gastro-intestinal loss due to diarrhea, heart failure, liver cirrhosis, nephrotic syndrome, medication-induced side effects, notably thiazide diuretics, and the syndrome of inappropriate antidiuresis (SIAD) [2].

SIAD is identified via urine osmolality of >100 mOsm/kg and urine sodium of >30 mmol/L after ruling out other potential causes, such as diuretic use, adrenal insufficiency, and hypothyroidism [3,4]. Notably, the concept of reset osmostat has emerged as a significant differential consideration in cases suggestive of SIAD [5,6]. Nevertheless, its recognition in clinical practice remains insufficient [7]. Initially reported in 1976, these cases were characterized by hypo-osmolar hypotonic hyponatremia with normal renal function, where sodium levels could not be normalized through water restriction or salt supplementation, primarily classified as SIAD type C [8]. Presently, reset osmostat is often considered a separate clinical phenomenon since this entails normal water load excretion and an intact urine dilution capability while exhibiting a normal sodium balance [5,9]. Although earlier publications show an ongoing debate regarding the nomenclature for reset osmostat, whether or not the terminology reset osmostat or SIAD type C is used, emphasis is made on the different mechanism compared to the ‘classical’ SIAD [10]. In this article, we consider reset osmostat as a separate entity. By distinguishing reset osmostat from SIAD, we aim to highlight the critical difference: reset osmostat exhibits normal free water excretion, unlike the decreased free water excretion observed in SIAD. This distinction is crucial when considering appropriate treatment approaches.

The literature regarding the prevalence of reset osmostat is scarce. Studies by Zerbe et al. (1980), Maesaka, and Chen et al. have reported incidences of 36%, 31%, and 62.5%, respectively, among individuals initially suspected of having SIAD. These study groups comprised patients from a general hospital ward and a long-term facility for the elderly [11,12,13]. Specific subgroup analyses, such as those involving tetraplegic patients and individuals with tuberculosis, also indicate a significant contribution of reset osmostat among individuals experiencing hyponatremia [14,15]. The water-diluting test currently serves as the primary diagnostic tool to distinguish reset osmostat from SIAD, with a notable differentiation in free water excretion rates (≥80% in reset osmostat versus 30–40% in SIAD) [5,16,17,18]. Additionally, normal uric acid excretion levels are considered indicative of reset osmostat [8,19]. The emerging role of copeptin, a byproduct of arginine vasopressin synthesis with a prolonged half-life, as a diagnostic marker is under investigation, though its definitive utility remains to be established [20].

In this study, we performed a prospective cohort study at a hospital ward and outpatient clinic to determine the frequency at which patients initially suspected of SIAD are diagnosed with reset osmostat using the water-diluting test as a gold standard. Furthermore, we explored the efficacy of alternative diagnostic tools, such as the 11% threshold of fractional uric acid excretion and copeptin.

## 2. Materials and Methods

The study was conducted in Isala, a large medical center in Zwolle, the Netherlands. This investigation initially aimed to be a prospective cohort study commencing in March 2021. Despite the screening of 763 cases, involving 711 unique patients, from March 2021 to March 2023, only 10 patients met the inclusion criteria, of whom 8 underwent the water-diluting test. Consequently, the study was prematurely concluded, leading to a modification in the research approach to a descriptive case series involving 8 patients. Patient selection was based on the presence of hypotonic hyponatremia, identified within 24 h of admission to the internal medicine ward or at the outpatient clinic. Hypotonic hyponatremia was defined according to serum sodium ≤ 130 mmol/L, serum osmolality ≤ 275 mOsmol/kg, urinary osmolality ≥ 100 mOsmol/kg, and urinary sodium ≥ 30 mmol/L, assessed through either spot or 24-h urine collections.

The study’s exclusion criteria encompassed manifest heart failure, decompensated liver cirrhosis, significant renal insufficiency (eGFR < 45 mL/min/1.73 m^2^ according to the CKD-EPI formula), untreated or inadequately managed hypothyroidism (TSH > 10 mU/L), and untreated or inadequately managed hypocortisolism (morning cortisol < 320 nmol/L). Additionally, patients on thiazide or loop diuretics, SGLT-2 inhibitors, or mannitol, those with isotonic or hypertonic hyponatremia (serum osmolality > 275 mOsmol/kg), urine osmolality < 100 mOsmol/kg, urine sodium of <30 mmol/L, obstructive bladder dysfunction (urine retention >200 cc after micturition), or osmotic diuresis (glucose > 15 mmol/L, calcium > 2.65 mmol/L), pregnant women, mentally incapacitated individuals, and those under the age of 18 years were excluded.

Eligible patients underwent a water-diluting test, adhering to standard care protocols, contingent upon serum sodium levels exceeding >120 mmol/L. After a modest breakfast with no more than 300 mL of water, patients were hospitalized. Throughout the test, they were instructed to stay seated or recumbent. Initially, they ingested 20 mL/kg of water within 30 min. The baseline and 4-h post-intake measurements included serum urea, creatinine, sodium, potassium, calcium, osmolality, uric acid, glucose, vasopressin, and copeptin; the 2-h assessments comprised sodium, osmolality, vasopressin, and copeptin. Urinary evaluations at the baseline and 4 h included creatinine, osmolality, sodium, urea, and uric acid, with an additional osmolality check at 2 h. Copeptin and vasopressin were sent to and analyzed in the laboratory of the University Medical Center Groningen (UMCG). In the case of a sodium drop of ≥4 mmol/L at the 2-h assessment, the test was preliminarily ended. A diagnosis of reset osmostat was established with ≥80% water intake excretion or a urinary osmolality of ≤100 mOsm/kg after 4 h. Indications of impaired water excretion, suggestive of SIAD, were identified via urinary osmolality ≥ 300 mOsm/kg with <80% water excretion, while results were deemed inconclusive if urinary osmolality ranged between >100 and <300 mOsm/kg with <80% water elimination.

## 3. Results

Over a two-year period, a water excretion test was conducted in eight patients, all of whom provided informed consent. The demographic details of these patients are summarized in Table 1.

Hyponatremia was predominantly chronic, having been present for several years in most cases, but it was confirmed at least 16 days prior to conducting the water-diluting test. The cohort predominantly comprised males aged 60 years or older with serum sodium of 123 to 130 mmol/L. Of the eight patients, five presented at the outpatient clinic, and the other three were hospitalized.

### 3.1. Primary Analysis

An exhaustive review of patient records was conducted to identify potential causes of hyponatremia. Factors like medication use, gastro-intestinal losses, and heart failure were ruled out. Fractional sodium excretion could not be determined for one patient, and it was less than 1% in the remaining seven patients. All patients had normal-to-high blood pressure and no clinical signs of peripheral edema or overfilling otherwise. The patients were, therefore, diagnosed with euvolemic hypotonic hyponatremia. Laboratory findings, detailed in Table 2 and Table 3, excluded conditions such as hypocortisolism, hypothyroidism, and osmotic diuresis, leaving SIAD, renal salt wasting, and reset osmostat as the remaining differential diagnoses.

### 3.2. Water-Diluting Test Results

The water-diluting test was completed by seven out of eight participants. Results form the test are shown in Table 4. For one individual, the test was prematurely terminated at 2 h due to a significant decrease in serum sodium levels from 127 mmol/L to 120 mmol/L, leading to a provisional diagnosis of SIAD. For the remaining seven patients, no significant drop in sodium levels was observed during the test, with a maximum difference of 2 mmol/L. All patients had initial sodium of ≥120 mmol/L.

For patients 1, 2, 3, and 8, the diagnosis of reset osmostat was confirmed based on water excretion of ≥80%. Remarkably, all these patients had fractional uric acid excretion of <11%. Patient 7 was predominantly diagnosed with SIAD, whereas the results for patients 4, 5, and 6 remained inconclusive. Variations in ADH levels did not correlate consistently with diagnoses, with ADH levels increasing in patient 1 and 2 and decreasing in patient 3 (despite a reset osmostat diagnosis), and copeptin levels decreasing across all the patients except patient 5, for whom this remained stable. The results did not reveal any notable differences between patients admitted to the ward and those visiting the outpatient clinic. Among the admitted patients, two were diagnosed with reset osmostat (patients 2 and 3) and one with SIAD (patient 7). For the outpatient clinic, three tests were inconclusive (patients 4, 5, and 6), while two patients had reset osmostat (patients 1 and 8).

For two of the four patients diagnosed with reset osmostat, no causative condition was identified. For patient 2, pancreatic carcinoma was deemed the probable cause. For patient 3, amitriptyline was implicated as the likely cause. For patient 4, the diagnosis of reset osmostat was primarily inferred from a significant water excretion rate and a corresponding decrease in urinary osmolality without identifying an underlying cause. For patient 5, SIAD was presumed to be induced via carbamazepine. For patient 6, the diagnosis of reset osmostat was supported by a longstanding history of hyponatremia and a notable reduction in urinary osmolality.

During the follow-up period, one patient diagnosed with reset osmostat passed away within three months due to pancreatic cancer. Among the remaining three reset osmostat patients, hyponatremia was left untreated and remained stable over time. A patient suspected of having SIAD was effectively treated with tolvaptan, resulting in normalized and steady serum sodium levels. Another patient with hyponatremia suspected to be caused due to carbamazepine continued taking the medication, resulting in stable but low serum sodium levels. The two remaining patients with possible SIAD were not treated and also remained steady during the follow-up period.

## 4. Discussion

The findings from this study underscore the rarity of reset osmostat as a diagnosis, evidenced by the selection of only eight patients over a two-year span following an exhaustive search. Identifying patients with reset osmostat among those suspected of having SIAD is of clinical significance, particularly because the standard practice of water restriction in SIAD management may not be necessary and could even be burdensome in cases of reset osmostat. While the rarity of this diagnosis may pose challenges in identifying such patients, our study suggests that analysis is particularly pertinent for individuals experiencing chronic hyponatremia. Given that this patient group had hyponatremia persisting for at least 16 days, a pragmatic and safe cutoff would be a duration of ≥7 days of hyponatremia for suspected SIAD patients, following the exclusion of other potential causes.

The prevailing view in the medical community is that reset osmostat typically does not warrant intervention, given the generally limited efficacy of treatment. However, this stance remains a subject of ongoing debate, especially considering the potential association between hyponatremia and adverse prognostic outcomes [5]. Chronic hyponatremia is not only associated with higher mortality rates but also sarcopenia, osteoporosis, and kidney stones [21]. Potential methods for increasing sodium levels by enhancing diuresis, as employed in SIAD, include oral urea or V2 receptor antagonists [22]. A study among chronic SIAD patients might provide insights into the effects of oral urea in reset osmostat as well. Unfortunately, this study does not specify whether some of the patients with chronic SIAD had reset osmostat. Considering the chronic nature of SIAD in these cases, it is likely that some patients in the study groups had reset osmostat, therefore suggesting a positive effect on the normalization of sodium in reset osmostat as well [23]. However, the relevance and especially the long-term effects of the normalization of sodium, particularly in reset osmostat, are unknown.

A broad analysis was conducted to rule out other diagnoses, as SIAD and reset osmostat are diagnoses of exclusion. Despite fractional sodium excretion being less than 1% in all patients, it is important to emphasize the high urine sodium levels and chronic nature of hyponatremia. Additionally, fractional sodium excretion is typically utilized in acute renal failure scenarios for the differentiation of prerenal causes or acute tubular necrosis, rendering it unreliable in cases of steady-state hyponatremia without clinical signs of hypovolemia [24]. Fractional sodium excretion might be helpful in predicting the effect of saline infusion [25]. Interestingly, in our study, the cases with reset osmostat exhibited a fractional sodium excretion of <0.5%, with one case having an unknown result. However, no effect is expected from saline infusion in reset osmostat. Future research could further investigate the relevance and implications of this finding. While the water-diluting test has historically been regarded as the optimal diagnostic approach for distinguishing SIAD from reset osmostat, recent studies have proposed that a threshold of 11% in fractional uric acid excretion could offer a more practical diagnostic criterion [26]. Among our study’s patients with reset osmostat, three out of four exhibited fractional uric acid excretion rates below this threshold. Such a measurement was not obtained for the fourth patient. This parameter was not applicable to our study’s patients with SIAD due to the limited number of diagnoses and the absence of fractional uric acid excretion data. Both diagnostic tools have their limitations. In the water dilution test, vasopressin escape may contribute to the decrease in urine osmolality with high water intake. Vasopressin escape is believed to be a mechanism that prevents severe hyponatremia in affected cases [27]. Fractional uric acid excretion may be less reliable in the elderly population. As a significant portion of suspected SIAD patients are elderly individuals, the relevance of this finding has yet to be established [28].

The inclusion of ADH and copeptin analysis in this study was motivated by the emerging recognition of copeptin as a potential diagnostic marker. A decrease in copeptin was expected in reset osmostat. Nevertheless, our findings did not corroborate the encouraging results reported in prior research [29]. A plausible reason for this discrepancy may be the limited number of patients in our cohort, coupled with an insufficient representation of SIAD for an effective comparison. Furthermore, the analysis of copeptin and ADH was conducted at another center, and the delay in testing and transport could have potentially influenced the results. Lastly, copeptin has been previously acknowledged as a marker with a wide overlap in various hyponatremia diagnoses and sensitivity to confounding factors such as emotional or physical stress prior to blood sample collection [30].

Following the identification of reset osmostat cases, it becomes imperative to investigate potential underlying etiologies, given the possibility that these conditions may necessitate medical intervention. The identification of causative factors in our study was constrained due to the small sample size, with inferences primarily drawn from individual case reports. Documented causes in the literature include intracranial arteriovenous malformations, intraventricular hemorrhage, cerebral palsy, Lewy Body dementia, colorectal and gastric carcinoma, venlafaxine use, and encephalitis [5,31,32,33,34,35,36,37].

Future research endeavors could involve conducting a multicenter study with a substantial cohort of patients. This could help solidify the utility of fractional uric acid excretion in diagnosing. Moreover, establishing a clearer set of patient characteristics, particularly regarding the duration of chronic hyponatremia, could enhance the understanding in this field.

## 5. Conclusions

This study reaffirms the rarity yet clinical significance of reset osmostat in the differential diagnosis of hyponatremia. Despite being rare, the chronic nature of suspected SIAD in patients makes it a comprehensive population deserving further analysis. The varied treatment approaches employed underscore the relevance of such analyses for patients. The water-diluting test, although instrumental in distinguishing reset osmostat from SIAD, is only suitable for a select group of patients. Our findings suggest that reset osmostat warrants consideration in instances of persistent hypotonic hyponatremia where other causes are not evident. Furthermore, fractional uric acid excretion below 11% emerges as a potentially valuable diagnostic marker in differentiating reset osmostat from other etiologies of hyponatremia.

## Figures and Tables

**Table 1 jcm-13-03538-t001:** Demographic data of study population.

Case	Gender	Age ^1^	Sodium ^1^	Duration Hyponatremia	Setting
1	Male	67	124	>10 years	Outpatient clinic
2	Male	83	123	16 days	Hospitalized
3	Female	81	127	±2 years	Hospitalized
4	Male	74	126	±1.5 years	Outpatient clinic
5	Female	60	126	75 days	Outpatient clinic
6	Male	71	130	±3 years	Outpatient clinic
7	Male	79	127	±7 years	Hospitalized
8	Male	62	129	±7 years	Outpatient clinic

^1^ Age in years; serum sodium in mmol/L.

**Table 2 jcm-13-03538-t002:** Baseline laboratory results.

Case	Osmol ^1^	Creat ^1^	eGFR ^1^	TSH ^1^	Cortisol ^1^	Glucose ^1^	Uric Acid ^1^	Potassium ^1^	Calcium ^1^
1	266	71	92	1.70	418	3.8	0.50	4.3	2.27
2	261	63	86	0.44	356	6.1	x ^3^	4.6	2.21
3	260	51	86	0.85	625	9.1	0.19	4.6	2.27
4	267	106	59	2.00	254 ^2^	7.9	0.18	4.3	2.21
5	267	44	109	1.00	659	5.5	0.12	4.3	2.12
6	271	70	90	1.40	384	5.6	0.22	4.4	x ^3^
7	265	67	86	1.50	404	6.6	0.21	4.5	2.30
8	270	47	113	3.50	360	9.1	0.18	4.8	x ^3^

^1^ Osmol in mOsm/kg; creatinine in µmol/L; eGFR in mL/min/1.73 m^2^ according to CKD-EPI formula; cortisol in nmol/L; glucose in mmol/L; uric acid in mmol/L; potassium in mmol/L; calcium in mmol/L corrected for albumin levels. ^2^ Hypocortisolism was excluded by performing a synacthen test. ^3^ Missing values.

**Table 3 jcm-13-03538-t003:** Baseline urinary results.

Case	Creatinine ^1^	Sodium ^1^	Osmol ^1^	Uric Acid ^1^	Fractional Uric Acid Excretion ^1^	Fractional Sodium Excretion
1	9.8	74	518	1.4	2	0.4
2	x ^2^	60	440	x ^2^	x ^2^	x ^2^
3	5.1	53	408	1.0	10.2	0.4
4	3.6	40	x ^2^	0.9	14	0.7
5	4.2	94	460	1.2	12	0.8
6	5.1	58	333	1.7	11	0.6
7	6.3	101	450	x ^2^	x ^2^	0.8
8	9.7	89	585	3.0	8	0.3

^1^ Creatinine in mmol/L; sodium in mmol/L; osmol in mOsm/kg; uric acid in mmol/L; fractional uric acid excretion in percentage (%); fractional sodium acid excretion in percentage (%). ^2^ Missing values.

**Table 4 jcm-13-03538-t004:** Water-diluting test results.

Case	Water Excretion ^1^	Urine Osmol (t = 0) ^1^	Urine Osmol (t = 4) ^1^	ADH (t = 0) ^1^	ADH (t = 4) ^1^	Copeptin (t = 0) ^1^	Copeptin (t = 4) ^1^
1	100%	458	230	1.5	1.8	13.4	8.9
2	100%	594	164	1.1	1.5	6.5	5.6
3	100%	414	166	0.71	0.64	6.0	3.8
4	70%	450	215	<0.25	<0.25	5.2	5.0
5	56%	475	269	<0.25	<0.25	4.0	4.0
6	32%	447	216	0.27	0.33	3.8	3.5
7	x ^2^	582	604 (t = 2)	1.0	0.81	8.9	7.7
8	121%	480	184	<0.25	<0.25	3.0	2.6

^1^ Water excretion: the amount of diuresis compared to water intake at the start of the test, expressed as a percentage (%); urine osmolality in mOsmol/kg at the start of the test (t = 0) and at the end of the test after 4 h (t = 4)—due to the early ending of the test for case 7, the result after 2 h is given in this case; ADH in ng/L at the start of the test (t = 0) and at the end of the test after 4 h (t = 4); copeptin in pmol/L at the start of the test (t = 0) and at the end of the test after 4 h (t = 4). ^2^ Missing value.

## Data Availability

The detailed raw data are not publicly available to preserve individuals’ privacy under the European General Data Protection Regulation. The authors agree to make data and materials supporting the results or analyses presented in their paper available upon reasonable request.

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
