# Peer review of "Chronic Hyponatremia: The Role of Reset Osmostat in Patients with Suspected SIAD"

_jcm, 2024, doi:10.3390/jcm13123538_

Round 1

Reviewer 1 Report

Comments and Suggestions for Authors

In this study, Kiewiet and co-authors have investigated patients with chronic hyponatremia and a SIADH-like appearance. They found that in such patients reset-osmostat is the likely pathology in approximately half of the patients according to the results of a water stress test. Moreover, they conclude that copeptin is not informative with regard to making the diagnosis of reset osmostat while FE of uric acid might be.

The authors have to be commended for their efforts to search for appropriate patients to include in their trial. In essence, they primarly prove that reset osmostat is an extremely rare condition. Still it exists and physicians should be aware of this often mild hyponatremia since it might be a hint to other underlying conditions.

From the numbers (FE uric acid, uric acid in serum and urine, urine creatinine) in the manuscript, the serum creatinine and then, in turn, FENa can be calculated in 6 of the 8 patients (since diuretic use was an exclusion criterion the FENa provides a valid insight into volume regulation). In all cases FENa was <1%, in two patients (1, 8) that were diagnosed with reset osmostat the FENa was even below 0.5%. This might indicate subclinical hypovolemia which is a known situation in which ADH release threshold can be resetted downwards. I would encourage the authors to include this piece of information in their discussion. Also, serum sodium levels should before and after water ingestion should be provided.

The small sample size precludes the authors from making more statistically sound conclusions. They emphasize this shortcoming correctly. Nonetheless, I do think that this work should be published to remind physicians of the existence of reset osmostat.

Author Response

Thank you very much for your kind feedback. We have added the fractional sodium excretion values in the new manuscript and further argumented our thoughts in the discussion, namely:

Despite fractional sodium excretion being less than 1% in all patients, which could suggest hypovolemia, it is important to emphasize the high urine sodium levels and chronic nature of the hyponatremia. Additionally, fractional sodium excretion is typically utilized in acute renal failure scenarios for the differentiation of prerenal causes or acute tubular necrosis, rendering it unreliable in cases of steady-state hyponatremia without clinical signs of hypovolemia.

Hopefully, this will help the further understanding of our study. Please let us know if any questions remain.

Kind regards, Aline Kiewiet

Reviewer 2 Report

Comments and Suggestions for Authors

The data don't increase our knowledge about SIAD ,

Serum SNa (not blood)

Reset osmostat represent about 20 % of SIAD (type C)

Elderly have a higher FE Uric acid ...it decrease also with normalisation of SNa

(see Musch et all Utility and limitation....International Urology and nephrology 2001;32:475-493)

-It is not because FeUric is normal during hyponatremia that it will not decrease after correction of SNa ( It is reported in 20-30 %)

- Are the patients treated ?these patient could be treated like the other types of SIAD (for exemple by urea)

-The decrease in Urine osmolality after the water load could reflect simply AVP escape.. (this must be discussed)

-In recet osmostat we expect a decrease in copeptin in all the patients 

Author Response

Dear reviewer,

We kindly thank you for your feedback on our manuscript. We made changes to the manuscript. And hereby we provide a reaction on your review. Please see the attachment for our reaction.

With kind regards,

Aline Kiewiet

Reviewer 3 Report

Comments and Suggestions for Authors

Review of the Manuscript "Chronic hyponatremia: the role of Reset Osmostat in patients with suspected SIADH"

The objective of the study was to determine the frequency at which patients, initially suspected of SIADH, are diagnosed with reset osmostat using the water diluting test as a gold standard, and to explore the efficacy of alternative diagnostic tools, such as the 11% threshold of fractional uric acid excretion. The manuscript is well-written, with clear and concise language that enhances the accessibility of the content.

Having thoroughly examined the content, I would like to pose a few clarifying questions to better understand certain aspects of the study.

·         You mentioned that in your experience copeptin’s relevance was not confirmed. How do you comment then the emerging interest for this biomarker in the scientific community?

·         Reset osmostat is rare. How would you test it then? Routinely or do you have any specific criteria for the patients who should be tested?

·         How could your research influence future research?

·         How could your findings influence clinical practice?

Kindly incorporate the responses within the manuscript to augment its overall quality.

Author Response

Dear reviewer,

We kindly thank you for your feedback. We made adjustments to the manuscript based on your feedback. Please see the attachment for our reaction on your questions. Please let us know if it remains unclear or you have further questions. Thank you very much.

With kind regards,

Aline Kiewiet

Round 2

Reviewer 2 Report

Comments and Suggestions for Authors

I disagree with the notion that recet osmostat is not a subtype of SIAD .These patients could be treated by urea for exemple (the urine osmolality will increase with the normalisation of SNa,Europ J Intern Med 2018)) Treatment of these patients must be done (decrease the Morbidity of chronic hyponatremia (J Clin Med 2023)

Introduction:Cirrhosis must be added

Hyponatremia can increase FeURIC by itself (Am J Kidney Disease 2000;36(4) 745-751

In reset osmostat treatment with urea increase urine osmolality to avoid hypernatremia (europ j intern med 2018)

Around 30 % of SIAD have a normal Fe uric of 10% but correction of hyponatremia is associated with a decrease of FeURIC (for exemple a patient

with siad  and a FeURIC of 10% during hyponatremia ,correction of SNa will be associated with a Fe uric of 7 % and uric acid will increase from 5 mgr/dl to 6,5 mgr/dl

Mild hyponatremia is associated with high morbidity (J Clin Med 2023)

In Siad FeNa is usualy higher than 0,5% (not 1%)Some patients with low FeNa <0,5%(reflecting low salt intake)could suggest salt depletion but these patients have usualy a high FE urea (see Musch et al Am J Med )

Author Response

Dear reviewer,

Thank you for your feedback and discussion points. We made some additional changes to our manuscript thanks to your feedback. In the attachment you can see our response to your feedback. We hope this improves our manuscript. Please let us know if any questions or remarks remain. 

Kind regards,

Aline Kiewiet, on behalf of the study team
